# Serine is the major residue for ADP-ribosylation upon DNA damage

**Luca Palazzo[1†], Orsolya Leidecker[2†], Evgeniia Prokhorova[1†], Helen Dauben[2], Ivan Matic[2*], Ivan Ahel[1*]**

[1]Sir William Dunn School of Pathology, University of Oxford, Oxford, United Kingdom; [2]Max Planck Institute for Biology of Ageing, Cologne, Germany

**Abstract** Poly(ADP-ribose) polymerases (PARPs) are a family of enzymes that synthesise ADP-ribosylation (ADPr), a reversible modification of proteins that regulates many different cellular processes. Several mammalian PARPs are known to regulate the DNA damage response, but it is not clear which amino acids in proteins are the primary ADPr targets. Previously, we reported that ARH3 reverses the newly discovered type of ADPr (ADPr on serine residues; Ser-ADPr) and developed tools to analyse this modification (Fontana et al., 2017). Here, we show that Ser-ADPr represents the major fraction of ADPr synthesised after DNA damage in mammalian cells and that globally Ser-ADPr is dependent on HPF1, PARP1 and ARH3. In the absence of HPF1, glutamate/aspartate becomes the main target residues for ADPr. Furthermore, we describe a method for site-specific validation of serine ADP-ribosylated substrates in cells. Our study establishes serine as the primary form of ADPr in DNA damage signalling.

DOI: https://doi.org/10.7554/eLife.34334.001

**\*For correspondence:**
imatic@age.mpg.de (IM);
ivan.ahel@path.ox.ac.uk (IA)

[†]These authors contributed equally to this work

**Competing interests:** The authors declare that no competing interests exist.

## Introduction

ADP-ribosylation (ADPr) is a reversible evolutionarily conserved posttranslational modification of proteins, which controls many critical cellular processes (*Palazzo et al., 2017a*; *Lüscher et al., 2018*).

Poly(ADP-ribose) polymerases (PARPs) compose the major family of enzymes that catalyse the transfer of ADP-ribose unit(s) from $NAD^+$ to protein substrates (*Barkauskaite et al., 2015*; *Gupte et al., 2017*). Seventeen members of the PARP superfamily are encoded within the human genome, which are characterized by distinct structural domains, activities and involvement in a variety of cellular processes, including the DNA damage response (DDR) (*Gupte et al., 2017*). PARPs directly involved in DNA repair are PARP1, PARP2, and PARP3 (*Langelier and Pascal, 2013*; *Martin-Hernandez et al., 2017*).

While PARP3 can attach only a single ADP-ribose unit on target proteins (MARylation) (*Vyas et al., 2014*), PARP1 and PARP2 can extend the initial ADPr event into long chains that remain attached on the proteins (Poly(ADP-ribosyl)ation, PARylation) (*D'Amours et al., 1999*). PARP-dependent ADPr of these proteins is induced by binding of PARPs to DNA breaks, which produces timely and localised ADPr signals that control appropriate DDR mechanisms (*Langelier et al., 2014*).

PARPs have previously been described to mainly target acidic residues (glutamates and aspartates; Glu and Asp, respectively) in proteins (*Gagné et al., 2015*; *Daniels et al., 2015*; *Martello et al., 2016*; *Crawford et al., 2018*). However, we recently showed that serine (Ser) residues are also targets for PARP-dependent protein modification (*Leidecker et al., 2016*; *Crawford et al., 2018*) and that Ser ADP-ribosylation (Ser-ADPr) is involved in processes underlying genome stability and the DDR, in particular (*Bonfiglio et al., 2017a*). Furthermore, we showed that the DNA damage responsive protein Histone PARylation Factor-1 (HPF1/C4orf27) forms complexes with either PARP1 or PARP2 (*Gibbs-Seymour et al., 2016*) and promotes the synthesis of Ser-ADPr

on a variety of protein substrates (*Bonfiglio et al., 2017a*). Finally, we discovered ARH3/ADPRHL2 as a hydrolase responsible for the specific reversal of Ser-ADPr in cells (*Fontana et al., 2017*).

Our previous observations suggested that Ser-ADPr is a widespread form of ADPr in cells (*Fontana et al., 2017*; *Bonfiglio et al., 2017a*). Here, by using a combination of biochemical and cell biology approaches, we demonstrate that the bulk of ADPr synthesised in cultured mammalian cells is strictly dependent on HPF1 and that Ser-ADPr represents the most abundant form of ADPr after DNA damage in these cells.

## Results and discussion

ADPr of proteins rapidly occurs to recruit and control activities of many crucial proteins involved in the repair of damaged DNA (*Martin-Hernandez et al., 2017*). The study of ADPr has been significantly hampered by technical limitations, such as the barriers to visualize all forms of cellular ADPr and the challenges in proteomics analyses (*Vivelo and Leung, 2015*; *Bonfiglio et al., 2017b*). For example, until recently, only anti-PAR antibodies have been available, which can detect only the long PAR chains. However, recently a reagent specific for ADPr of any length (referred here as a pan-ADPr antibody) as well as a reagent specific for mono-ADPr have been developed (*Gibson et al., 2017*) and allowed us to follow protein ADPr events in cells upon DNA damage. We first exposed human osteosarcoma U2OS to the DNA damaging agent hydrogen peroxide ($H_2O_2$) and compared the ADPr pattern of control, ARH3 knock-out (KO), HPF1 KO and PARP1 KO cells (*Figure 1A*). In control cells, pan-ADPr signals after DNA damage revealed a number of modified proteins in the extract. The most easily identifiable signals relate to the modification of histone proteins as well as PARP automodification (*Figure 1A*). Both signals can also be recognised by the reagent that is specific for MARylation (referred here as a mono-ADPr antibody; *Figure 1A*) (*Gibson et al., 2017*). 2 hr after DNA damage the global ADPr signal is reduced to the levels comparable to untreated cells. However, importantly, the DNA damage-inducible ADPr is prevented in HPF1-depleted cells (*Figure 1A*), as we observed previously for specific histone substrates (*Bonfiglio et al., 2017a*). An exception is the auto-modification of PARP1 that is characterized by relatively longer ADPr chains (although also at overall lower levels) when compared to control cells (*Figure 1A*), as noted previously (*Gibbs-Seymour et al., 2016*). Expectedly, most of the global ADPr signal was abolished in PARP1 KO cells, as we showed previously for histone Ser-ADPr marks (*Bonfiglio et al., 2017a*), confirming that PARP1 is the most active PARP involved in DDR (*Figure 1A*). In order to investigate whether the global pan-ADPr is truly dependent on HPF1, we tested two independent clones of HPF1 KO cells and observed comparable results with both cell lines (*Figure 1B*). These data suggest that global ADPr in response to DNA damage requires both HPF1 and PARP1.

As observed in our previous study (*Fontana et al., 2017*), ARH3 KO cells already showed notably higher levels of ADPr proteins under unstimulated conditions when compared to control cells (*Figure 1A–B*). However, the difference in ADPr signal upon DNA damage was much more pronounced and allowed detection of a number of additional ADP-ribosylated protein bands in ARH3-deficient extracts that persisted for at least 2 hr. This was especially obvious for histone ADPr detected by the pan-ADPr reagent (*Figure 1A–B*).

Next, we used another DNA damaging agent, the alkylating agent methyl methanesulfonate (MMS). U2OS cells were treated with MMS, and then their recovery was analysed at 40' and 120' time points. Consistent with the observations for $H_2O_2$ treatment, most of the ADPr signal was HPF1-dependent and persisted in ARH3-deficient cells (*Figure 1C*).

In order to validate our observations in other cellular models, we next tested the human embryonic kidney (HEK) 293 cells. To confirm whether ADPr is HPF1-dependent in these cells, we challenged wild type and HPF1-depleted HEK293 cells with $H_2O_2$ (*Figure 1D*). Notably, HEK293 cells showed a number of ADP-ribosylated proteins detected by pan-ADPr antibody and modifications of most of these proteins were strictly dependent on HPF1 protein (*Figure 1D*).

The dependence of ADPr regulation on HPF1 and ARH3 suggests that Ser-ADPr is the dominant form of ADPr upon DNA damage. This is consistent with the data from recent proteomic analyses capable of detecting Ser-ADPr (*Leidecker et al., 2016*; *Bonfiglio et al., 2017a*; *Bilan et al., 2017*).

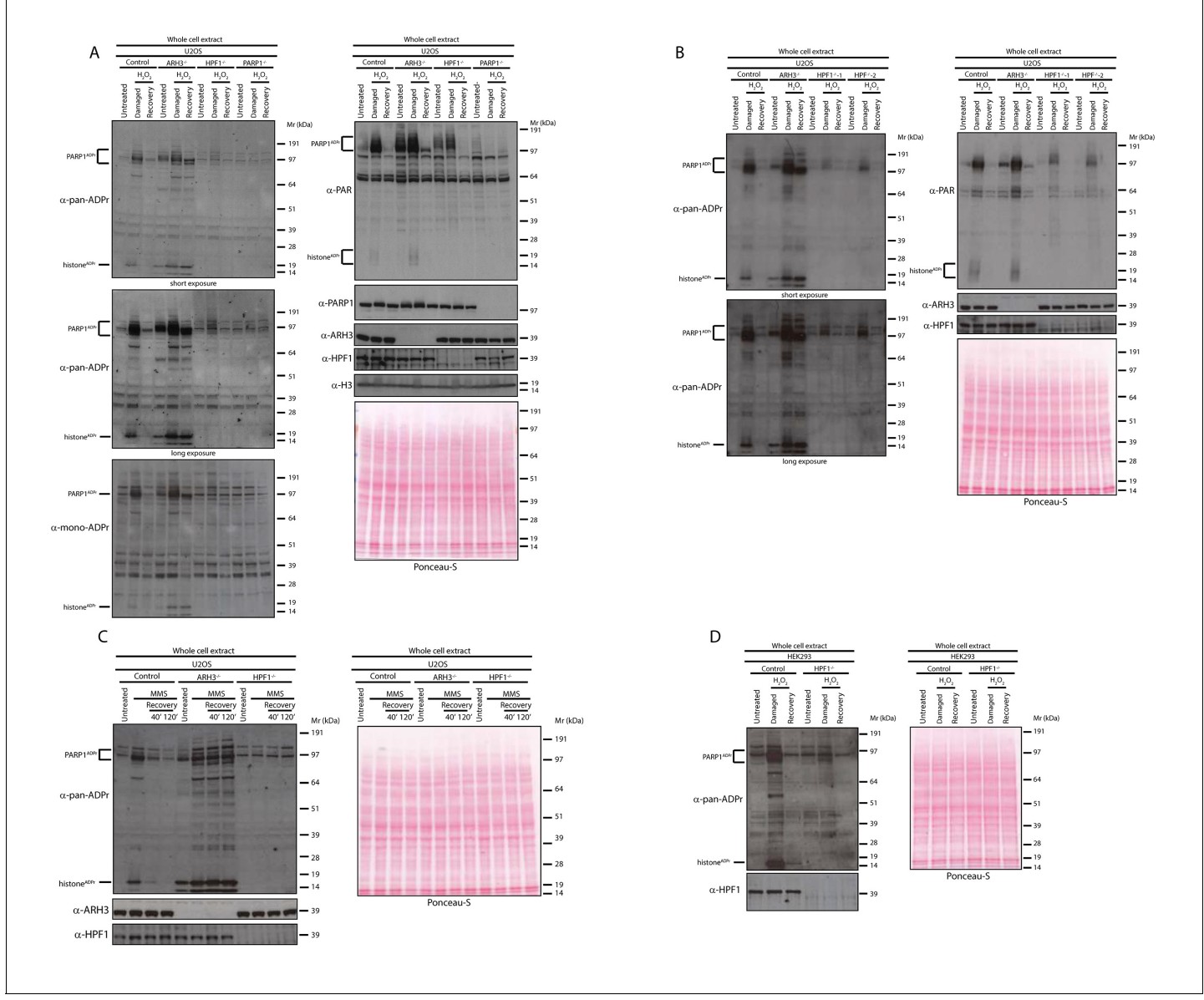

**Figure 1.** HPF1-dependent Ser-ADPr is the major form of ADPr upon genotoxic stress. (**A**) Control, ARH3 KO (ARH3$^{-/-}$), HPF1 KO (HPF1$^{-/-}$), and PARP1 KO (PARP1$^{-/-}$) U2OS cells were treated with 2 mM $H_2O_2$. After treatment/recovery, cells were lysed and proteins were separated by SDS-PAGE, analysed by western blot and probed for pan-ADPr, mono-ADPr, PAR, PARP1, ARH3, H3, and HPF1 antibodies. Additionally, Ponceau-S staining was used as loading control. (**B**) Control, ARH3 KO (ARH3$^{-/-}$) and two independent clones of HPF1 KO (HPF1$^{-/-}$−1 and HPF1$^{-/-}$−2) U2OS cells were treated with 2 mM $H_2O_2$. After treatment/recovery, cells were lysed and proteins were separated by SDS-PAGE, analysed by western blot and probed for pan-ADPr, PAR, ARH3, and HPF1 antibodies. Ponceau-S staining was used as loading control. (**C**) Control, ARH3 KO (ARH3$^{-/-}$), and HPF1 KO (HPF1$^{-/-}$) U2OS cells were treated with 2 mM MMS. After the induction of DNA damage, the cells were left to recover from genotoxic stress for the indicated time points. After treatment/recovery, cells were lysed and proteins were separated by SDS-PAGE, analysed by western blot and probed for pan-ADPr, ARH3, and HPF1 antibodies. Ponceau-S staining was used as loading control. (**D**) Control and HPF1 KO (HPF1$^{-/-}$) HEK293 cells were treated with 2 mM $H_2O_2$. After treatment/recovery, cells were lysed and proteins were separated by SDS-PAGE, analysed by western blot and probed for pan-ADPr, ARH3, and HPF1 antibodies. Ponceau-S staining was used as loading control.

DOI: https://doi.org/10.7554/eLife.34334.002

However, proteomics studies based on hydroxylamine, which excludes ADPr mapping on residues other than the Glu and Asp (*Moss et al., 1983*; *Daniels et al., 2015*), showed that modification of these residues is widespread in DDR (*Zhang et al., 2013*; *Gibson et al., 2017*; *Zhen et al., 2017*). We reasoned that hydroxylamine could allow a direct and simple estimate of the abundance of the

modification on Glu/Asp in the context of the global ADPr. Importantly, using defined substrates for both Ser-ADPr (*Bonfiglio et al., 2017a*; *Fontana et al., 2017*) and Glu/Asp-ADPr (*Sharifi et al., 2013*), we showed by autoradiography that hydroxylamine does not remove ADPr from Ser residues (*Figure 2A*) while confirming the complete removal of ADPr from Glu/Asp (*Figure 2B*). Considering the above data (*Figure 1*) and previous studies (*Bilan et al., 2017*; *Leidecker et al., 2016*; *Bonfiglio et al., 2017a*; *Fontana et al., 2017*) that imply the predominance of Ser-ADPr upon DNA damage, we hypothesized that the loss of signal after hydroxylamine treatment would be minor in wild type cells. To test this, we incubated hydroxylamine with proteins extracted under denaturing conditions from both control (*Figure 2C*) and H$_2$O$_2$–treated (*Figure 2D*) cells and monitored the ADPr signal with the anti-pan-ADPr and anti-PAR antibodies. We observed no noticeable loss of either the global ADPr or PARylation signal in DNA-damaged cells (*Figure 2D*) and only a moderate reduction in untreated cells (*Figure 2C*), indicating that under these conditions Glu/Asp-ADPr is not

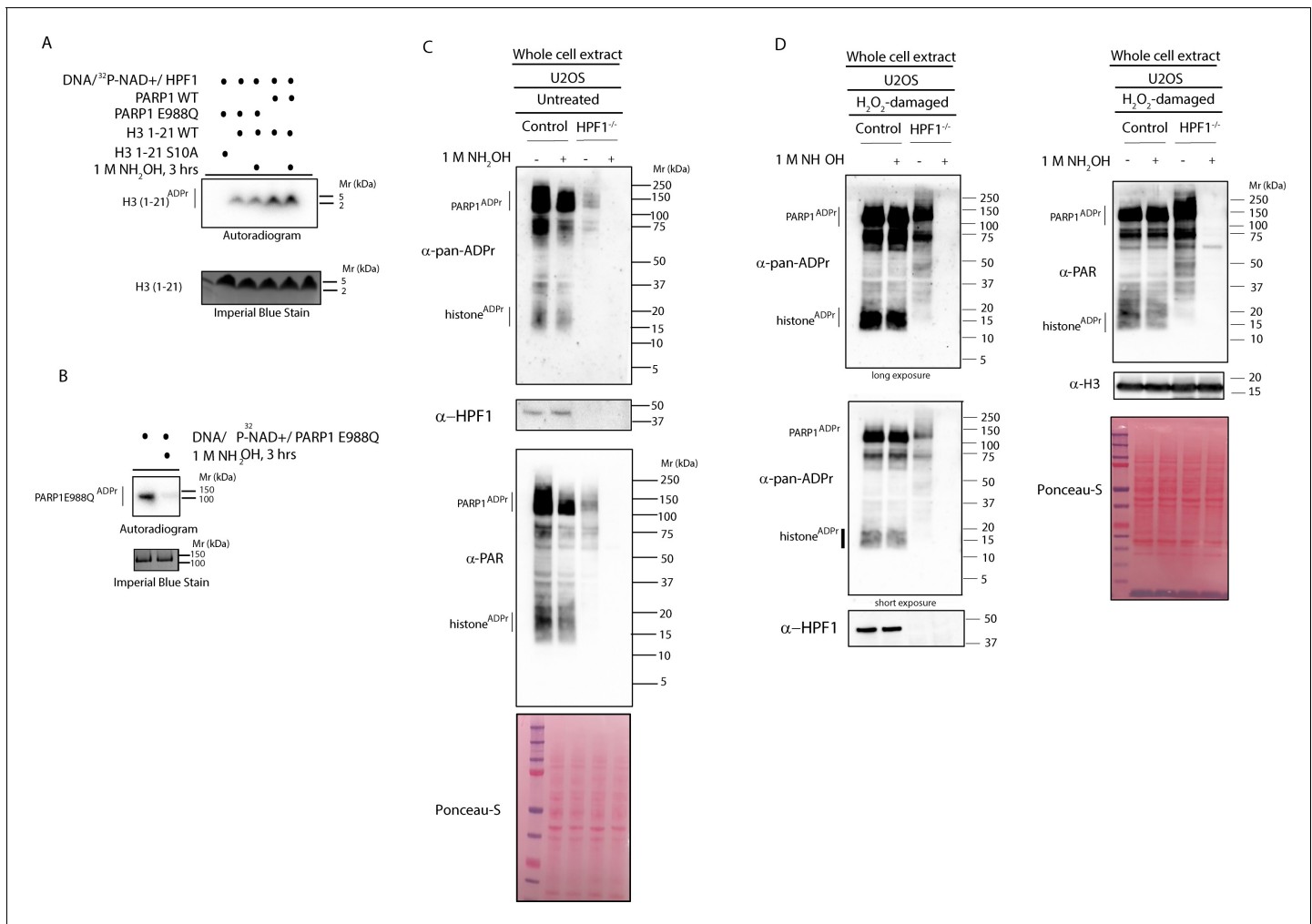

**Figure 2.** HPF1-dependent ADPr is resistant to hydroxylamine. (**A**) Autoradiogram shows serine ADPr of two synthetic peptides (wild type (WT) or Ser10Ala (S10A) mutant) corresponding to amino acids 1–21 of human H3 by wild type PARP1 or PARP1 E988Q in the presence of HPF1, with or without treatment with 1M NH$_2$OH (hydroxylamine). Imperial Blu staining was used to show equal loading of samples. (**B**) Autoradiogram shows auto-ADPr of PARP1 E988Q (at glutamate residues) and the effect of the treatment with 1M NH$_2$OH. Imperial Blue staining was used to show equal loading of samples. (**C**) Whole cell extracts were prepared from pre-damaged U2OS wild type or HPF1 KO (HPF1$^{-/-}$) cells. Extracts were either left untreated or treated with 1M hydroxylamine (NH$_2$OH) for 3 hr prior to separation on SDS-PAGE gel and immunoblotting with pan-ADPr, PAR or HPF1 antibodies. Ponceau-S staining was used as loading control. (**D**) Whole cell extracts were prepared from U2OS wild type or HPF1 KO (HPF1$^{-/-}$) cells following treatment with 2 mM H$_2$O$_2$ for 10'. Extracts were either left untreated or treated with 1M hydroxylamine (NH$_2$OH) for 3 hr prior to separation on SDS-PAGE gel and immunoblotting with pan-ADPr, PAR, H3 or HPF1 antibodies. Ponceau-S staining was used as loading control.
DOI: https://doi.org/10.7554/eLife.34334.003

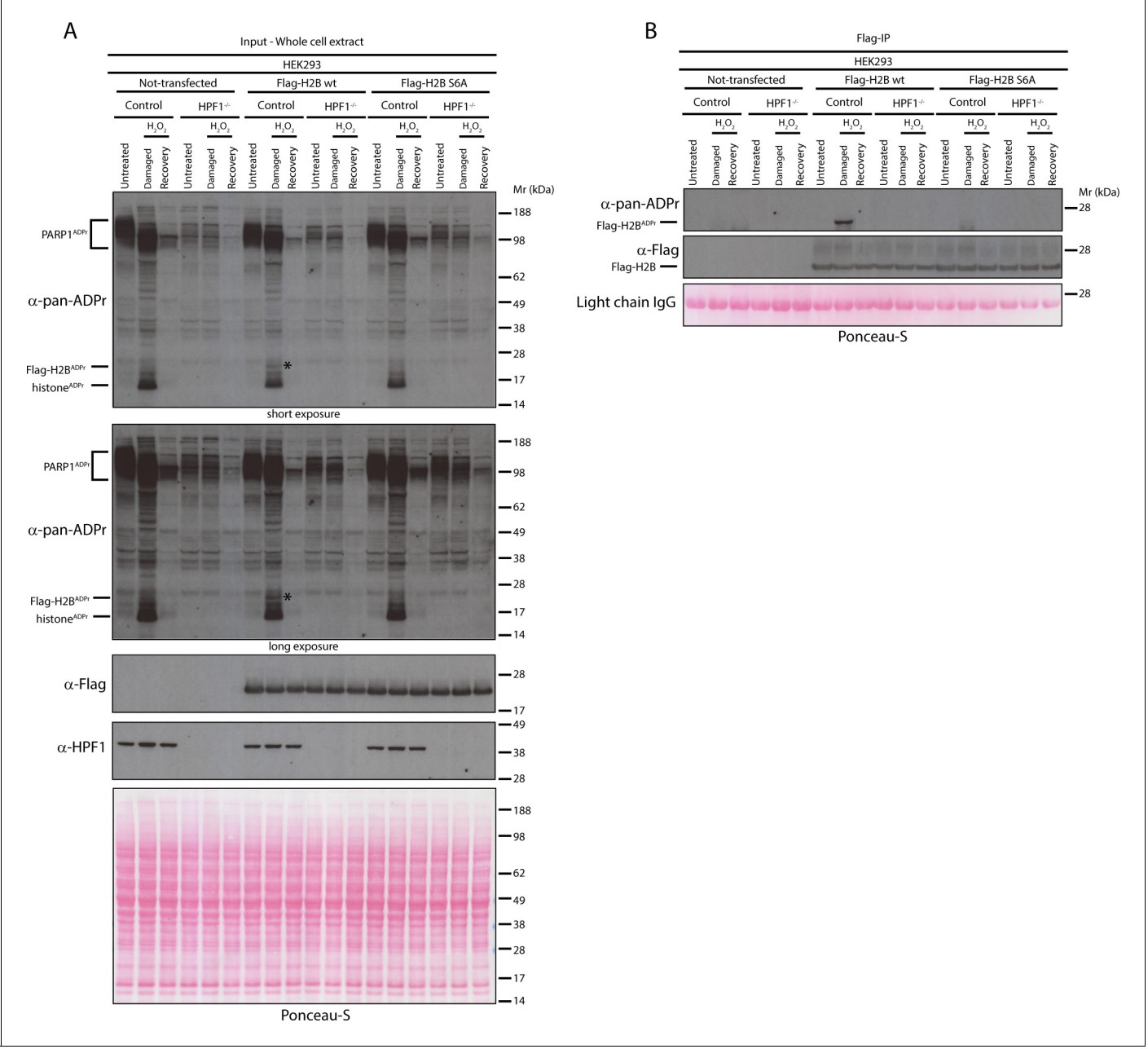

**Figure 3.** Serine six is the main ADPr site of histone H2B induced by DNA damage. (**A**) Control and HPF1 KO (HPF1$^{-/-}$) HEK293 cells were transfected or not with Flag-H2B wild type (wt) and Flag-H2B Ser6Ala mutant construct (S6A). 24 hr post-transfection, cells were treated with 2 mM H$_2$O$_2$. After treatment/recovery, cells were lysed and proteins were separated by SDS-PAGE, analysed by western blot and probed for pan-ADPr, Flag, and HPF1 antibodies. Ponceau-S staining was used as loading control. The black star marks the ADP-ribosylated Flag-tagged H2B protein in the whole cell wild type extract, which is absent in other extracts. (**B**) Flag-tagged H2B wild type (wt) and Ser6Ala mutant (S6A) were immunoprecipitated (IP) by using anti-Flag antibody from the lysates generated in *Figure 3A*. IPs were separated by SDS-PAGE, analysed by western blot and probed for pan-ADPr and Flag antibodies. Ponceau-S staining was used to stain light chains of Immunoglobulins (IgG) as loading control of the IP.
DOI: https://doi.org/10.7554/eLife.34334.004

abundant. In contrast, intriguingly, hydroxylamine completely abolished both the global ADPr and PAR signals on proteins extracted from HPF1 KO cells (*Figure 2C–D*), implying that in the absence of HPF1 virtually all the ADPr is on acidic residues. This cellular finding is in accordance with the well-established biochemical evidence that Asp and Glu are the prevalent target residues when in

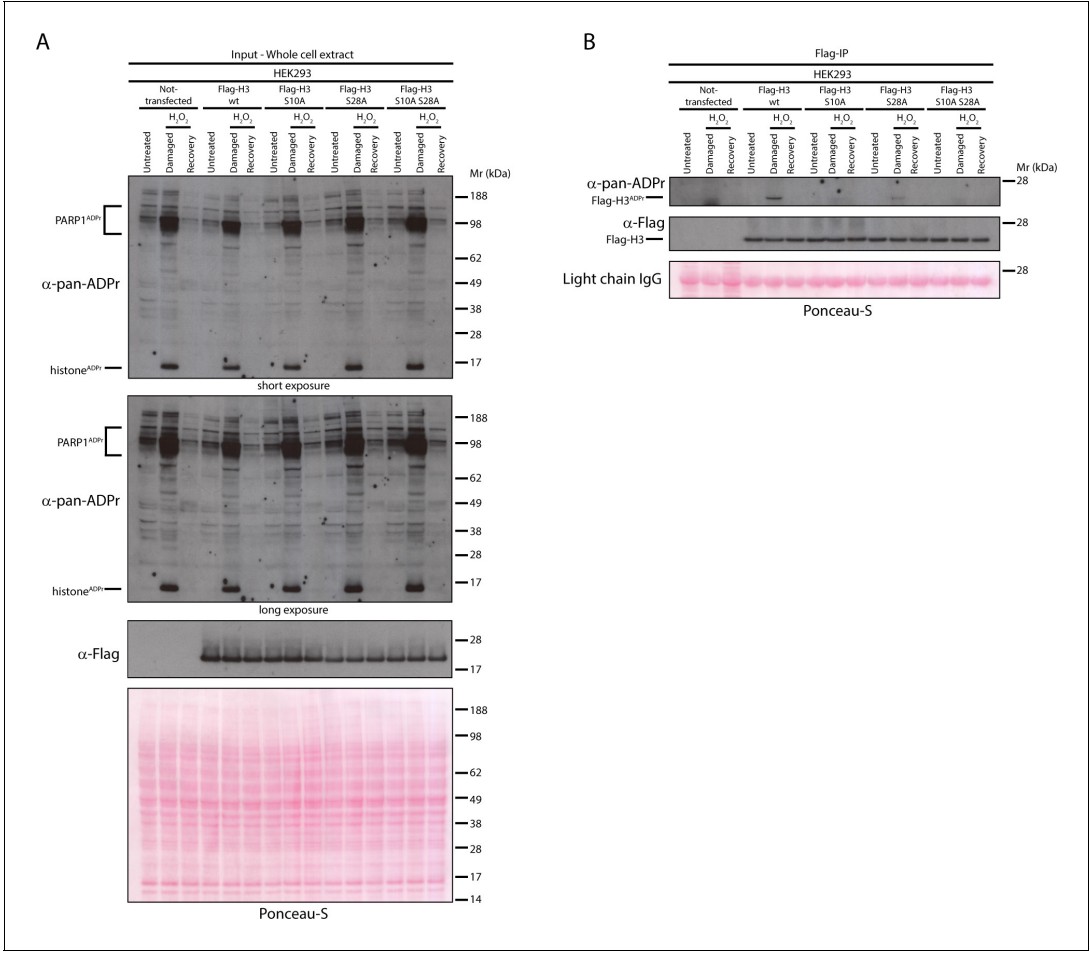

**Figure 4.** Serine 10 and serine 28 are the main ADPr sites of histone H3 induced by DNA damage. (**A**) HEK293 cells were transfected or not with Flag-H3.1 (Flag-H3) wild type (wt), Flag-H3.1 Ser10Ala (S10A), Flag-H3.1 Ser28Ala (S28A), and Flag-H3.1 Ser10Ala Ser28Ala double mutant (S10A S28A) constructs. 24 hr post-transfection, cells were treated with 2 mM $H_2O_2$. After treatment/recovery, cells were lysed and proteins were separated by SDS-PAGE, analysed by western blot and probed for pan-ADPr and Flag antibodies. Ponceau-S staining was used as loading control. (**B**) Flag-tagged H3.1 (Flag-H3) wild type (wt), Ser10Ala (S10A), Ser28Ala (S28A), and Ser10Ala Ser28Ala double mutants (S10A S28A) were immunoprecipitated (IP) by using anti-Flag antibody from the lysates generated in *Figure 4A*. IPs were separated by SDS-PAGE, analysed by western blot and probed for pan-ADPr and Flag antibodies. Ponceau-S staining was used to stain light chains of immunoglobulins (IgG) as loading control of the IP.

DOI: https://doi.org/10.7554/eLife.34334.005

vitro ADPr reactions are performed without HPF1 (*Tao et al., 2009*; *Sharifi et al., 2013*; *Bonfiglio et al., 2017a*).

The above experiments demonstrate the abundance of Ser-ADPr by monitoring global ADPr level (*Figure 1*). To investigate this on a specific substrate, we analysed the level of Ser-ADPr on histone H2B. In our previous studies, we identified by mass spectrometry ADPr of H2B exclusively on Ser residues (*Leidecker et al., 2016*) and showed that this modification is highly dependent on HPF1 in cells (*Bonfiglio et al., 2017a*). Here, we set to investigate whether Ser-ADPr is the primary form of ADPr on histone H2B. To test this, we generated mammalian expression constructs where H2B was tagged with the Flag epitope as well as a version of Flag-tagged H2B where the main candidate ADPr site, Ser6 was mutated to alanine (S6A). Both wild type and S6A Flag-tagged H2B constructs were transiently transfected into control and HPF1-depleted HEK293 cells. Whilst the Flag-tagged wild type H2B was efficiently ADP-ribosylated upon $H_2O_2$ treatment in control cells, the H2B ADPr signal was completely abolished in HPF1 KO HEK293 cells, as shown in both whole cell extracts and Flag-immunoprecipitations (IP) (*Figure 3A–B*, respectively). In addition, the S6A mutant did not show any ADPr in both control and HPF1-depleted cells. These data demonstrate that Ser residue at

position six is the main acceptor site of ADPr on H2B (*Leidecker et al., 2016*) and confirm that this Ser-ADPr is HPF1-dependent (*Bonfiglio et al., 2017a*).

Next, we used the same approach to confirm the main in vivo ADPr sites on histone H3. In our previous study, we detected H3 ADPr sites in cells on Ser10 and Ser28 (*Leidecker et al., 2016*), so we prepared the constructs for the expression of the Flag-tagged H3 wild type protein, the H3 alanine mutants at Ser10 (S10A) and Ser28 (S28A), as well as the double mutant (S10A S28A). Flag-IP of the Flag-tagged proteins expressed in HEK293 cells and subsequent western blot against pan-ADPr revealed that the mutation of both Ser10 and Ser28 on H3 abolishes the DNA damage-induced H3 ADPr (*Figure 4*). Furthermore, we observed that in our conditions ADPr predominantly happens on the Ser10 site, while the mutation of Ser28 showed a small, but significant reduction of ADPr. In conclusion, the expression of wild type and Ser-ADPr mutants in HEK293 cells by this simple approach should allow the validation of Ser-ADPr sites for many other ADP-ribosylated candidate proteins involved in the DDR.

ADPr is unique among posttranslational modifications for its exceptional chemical versatility in modifying a variety of substrate amino acids (*Daniels et al., 2015*). Glu, Asp, Lys, Arg and Ser have been indicated as the major ADPr target residues by recent proteomics studies, some of which, however, employ sub-optimal approaches that may lead to misassignment of ADPr specificities (*Bonfiglio et al., 2017b*). Thus, the attention is currently shifting from mere identification of ADPr sites to the elucidation of the biological pathways in which a form of ADPr plays a major role (*Gupte et al., 2017*). Our discovery of Ser-ADPr in 2016 has fuelled the rapid progress in the field, which has already resulted in the identification of the 'eraser' of Ser-ADPr (*Fontana et al., 2017*). Our computational reanalysis (*Matic et al., 2012*) of a published ADPr dataset (*Martello et al., 2016*); ProteomeXchange ID: PXD004245) showed that Ser-ADPr is a widespread modification (*Bonfiglio et al., 2017a*). This has stimulated the current study, in which we address the abundance of Ser-ADPr in cells. Our findings show that in endogenous wild type cells Ser-ADPr is the primary form of ADPr upon DNA damage. This is consistent with the data from an independent study, in which an unbiased proteomics technology (*Bonfiglio et al., 2017b*) showed that the vast majority of ADPr localises on Ser residues (*Bilan et al., 2017*). Importantly, our data indicate that Asp and Glu are the main targets of ADPr in PARP1-dependent, but HPF1-independent DNA damage signalling. This extends the concept of the 'switching' of PARP1 amino acid specificity (*Bonfiglio et al., 2017a*; *Leung, 2017*) to the cellular context. Future studies are needed to unravel the physiological and pathological conditions controlled by Ser-ADPr and Asp/Glu-ADPr.

In conclusion, the discovery of Ser-ADPr as well as the recent discoveries of reversible ADPr of DNA (*Jankevicius et al., 2016*; *Talhaoui et al., 2016*; *Munnur and Ahel, 2017*; *Dölle and Ziegler, 2017*) have added considerable depth to our understanding of the function and versatility of ADPr signalling in the cell, and raise the possibility that there may be other unique cellular and molecular processes regulated by ADPr.

## Materials and methods

**Key resources table**

| Reagent type (species) or resource | Designation | Source or reference | Identifiers | Additional information |
|---|---|---|---|---|
| cell line (Homo sapiens) | U2OS | ATCC | HTB-96, RRID:CVCL_0042 | |
| cell line (Homo sapiens) | HEK293 | ATCC | CRL-3216, RRID:CVCL_0063 | |
| cell line (Homo sapiens) | U2OS ARH3 KO | *Fontana et al., 2017* | | |
| cell line (Homo sapiens) | U2OS HPF1 KO | *Gibbs-Seymour et al. (2016)* | | |
| cell line (Homo sapiens) | U2OS PARP1 KO | *Gibbs-Seymour et al. (2016)* | | |
| cell line (Homo sapiens) | HEK293 HPF1 KO | *Gibbs-Seymour et al. (2016)* | | |
| antibody | anti-PAR (rabbit polyclonal) | Trevigen (Gaithersburg, MD, US) | 4336-BPC-100, RRID:AB_2721257 | WB 1:1000 |

*Continued on next page*

*Continued*

| Reagent type (species) or resource | Designation | Source or reference | Identifiers | Additional information |
|---|---|---|---|---|
| antibody | anti-pan-ADP-ribose (rabbit monoclonal) | Millipore (Billerica, MA, US ) | MABE1016, RRID:AB_2665466 | WB 1:1500 |
| antibody | anti-mono-ADP-ribose (rabbit monoclonal) | Millipore (Billerica, MA, US ) | MABE1076, RRID:AB_2665469 | WB 1:1000 |
| antibody | anti-PARP1 [E102] (rabbit monoclonal) | Abcam (Cambridge, UK) | ab32138, RRID:AB_777101 | WB 1:1000 |
| antibody | anti-histone H3, CT, pan (rabbit polyclonal) | Millipore (Billerica, MA, US ) | 07–690, RRID:AB_417398 | WB 1:2000 |
| antibody | anti-ARH3/ADPRH (rabbit | Atlas Antibodies (Stockholm, Sweden) | HPA027104, RRID:AB_10601330 | WB 1:1000 |
| antibody | anti-HPF1 (rabbit polyclonal) | *Gibbs-Seymour et al. (2016)* | | WB 1:1000 |
| antibody | anti-Flag HRP-conjugated (mouse monoclonal) | Sigma-Aldrich (St. Louis, MO, US) | A8592, RRID:AB_439702 | WB 1:5000 |
| antibody | anti-Flag M2 agarose-conjugated (mouse monoclonal) | Sigma-Aldrich (St. Louis, MO, US) | A2220, RRID:AB_10063035 | IP |
| recombinant DNA reagent | pDONR221 (Gateway vector) | Thermo Fisher Scientific (Waltham, MA, US) | 12536017 | |
| recombinant DNA reagent | pDEST C3X (Gateway vector) | other | | Laboratory of Fumiko Esashi |
| recombinant DNA reagent | Flag-H2B wt (plasmid) | This paper | | Progentiors: pDONR221-H2B; Gateway vector:pDEST C3X |
| recombinant DNA reagent | Flag-H3.1 wt (plasmid) | This paper | | Progentiors: pDONR221-H3.1; Gateway vector:pDEST C3X |
| recombinant DNA reagent | Flag-H2B S6A (plasmid) | This paper | | Made from Flag-H2B wt by site-directed mutagenesis |
| recombinant DNA reagent | Flag-H3.1 S10A (plasmid) | This paper | | Made from Flag-H3.1 wt by site-directed mutagenesis |
| recombinant DNA reagent | Flag-H3.1 S28A (plasmid) | This paper | | Made from Flag-H3.1 wt by site-directed mutagenesis |
| recombinant DNA reagent | Flag-H3.1 S10A S28A (plasmid) | This paper | | Made from Flag-H3.1 S10A by site-directed mutagenesis |
| peptide, recombinant protein | Human PARP1 | Trevigen (Gaithersburg, MD, US) | 4668–02 K-01 | |
| peptide, recombinant protein | Human PARP1 E988Q | *Fontana et al., 2017* | | |
| peptide, recombinant protein | Human HPF1 | *Gibbs-Seymour et al. (2016)* | | |
| peptide, recombinant protein | Human histone H3 fragment (1-21) wt | *Bonfiglio et al., 2017a* | | |
| peptide, recombinant protein | Human histone H3 fragment (1-21) S10A | *Bonfiglio et al., 2017a* | | |
| chemical compound, drug | Olaparib | Cayman Chemical (Ann Arbor, MI) | 10621 | |
| chemical compound, drug | ADP-HPD, dihydrate, ammonium salt | Calbiochem (La Jolla, CA) | 118415 | |
| chemical compound, drug | Hydrogen peroxide | Sigma-Aldrich (St. Louis, MO, US) | H1009 | |
| chemical compound, drug | Methyl methanesulfonate | Sigma-Aldrich (St. Louis, MO, US) | 129925 | |
| chemical compound, drug | Hydroxilamine | Sigma-Aldrich (St. Louis, MO, US) | 438227 | |

## Antibodies

Anti-PAR polyclonal antibody (4336-BPC-100, RRID:AB_2721257; rabbit) was purchased from Trevigen (Gaithersburg, MD , U S ) and used at 1:1000 dilutions. Monoclonal anti-pan-ADPr (MABE1016, RRID:AB_2665466), monoclonal anti-mono-ADPr (MABE1076, RRID:AB_2665469) and polyclonal

anti-histone H3 (07–690, RRID:AB_417398) rabbit antibodies were purchased from Millipore (Biller-ica, MA, US ) and used at 1:1500, 1:1000 and 1:2000 dilutions, respectively. Rabbit polyclonal anti-ARH3/ADPRHL2 (HPA027104, RRID:AB_10601330) was purchased from Atlas Antibodies (Stock-holm, Sweden) and used at 1:1000 dilution. Rabbit monoclonal anti-PARP1 (ab32138, RRID:AB_777101) was purchased from Abcam (Cambridge, UK) and used at 1:1000 dilution. Custom-made rabbit polyclonal HPF1 antibody was used as described (1:1000) (*Gibbs-Seymour et al., 2016*). Anti-Flag M2 agarose affinity gel (A2220, RRID:AB_10063035) and anti-Flag HRP-conjugated (A8592, RRID:AB_439702; used at 1:5000 dilution) mouse monoclonal antibodies were purchased from Sigma-Aldrich (St. Louis, MO, US). Immunoblots were performed as previously described (*Fontana et al., 2017*).

## Cell lines
Human U2OS osteosarcoma (ATCC HTB-96, RRID:CVCL_0042) and HEK293 (ATCC CRL-3216, RRID: CVCL_0063) cells were acquired from ATCC, identity was confirmed by STR profiling, and absence of mycoplasma contamination confirmed by MycoAlert Mycoplasma Detection Kit. Cells were cul-tured as previously described (*Fontana et al., 2017*). Generation of KO cell lines was previously described (*Gibbs-Seymour et al., 2016*; *Fontana et al., 2017*).

## Plasmid constructs
Full-length human histones H2B and H3.1 cDNA were cloned into the pDONR221 vector (Thermo Fisher Scientific; Waltham, MA, US). Ser to Ala point mutations were produced in pDONR221-H2B and pDONR-H3.1 by site directed mutagenesis. Mammalian expression constructs expressed H2B and H3.1 proteins with the C-terminal 3xFlag tag.

## Transfection
Transient DNA transfections in HEK293 cells were performed with Polyfect (QIAGEN; Venlo, Nether-lands) for 24 hr.

## Induction of DNA damage, preparation of cell extracts
For MMS treatment, cells were damaged with 2 mM MMS (Sigma-Aldrich; St. Louis, MO, US) for 1 hr. In case of $H_2O_2$, cells were damaged with 2 mM $H_2O_2$ (Sigma-Aldrich; St. Louis, MO, US) for 10 min. Cells were lysed as previously described (*Fontana et al., 2017*) in the following buffer: 50 mM Tris-HCl pH 8.0, 100 mM NaCl, and 1% Triton X-100. Immediately before lysing the cells, the lysis buffer was supplemented with 5 mM $MgCl_2$, 1 mM DTT, proteases and phosphatases inhibitors (Roche; Basel, Switzerland), 1 μM ADP-HPD (Calbiochem, La Jolla, CA), and 1 μM Olaparib (Cayman Chemical, Ann Arbor, MI). After the cell pellet was resuspended in the supplemented lysis buffer, Benzonase (Sigma-Aldrich; St. Louis, MO, US) was added (*Fontana et al., 2017*).

## Hydroxylamine experiments
Cell pellets were resuspended in SDS lysis buffer (10 mM HEPES pH 8.0, 2 mM $MgCl_2$, 1% SDS, 250 U Universal Nuclease (Pierce; Waltham, MA, US), 1 x protease inhibitor (Roche; Basel, Switzerland) and briefly sonicated. BCA assay (Pierce; Waltham, MA, US) was used to determine the protein con-centration. 30 μg damaged or 50 μg non-damaged cell were treated with 1 M $NH_2OH$ (hydroxyl-amine; Sigma-Aldrich; St. Louis, MO, US) for 3 hr at room temperature or left untreated. After the treatment, extracts were neutralized with 0.3% HCl and mixed with 4x SDS Loading buffer (Invitro-gen; Calrsbad, CA, US) containing 100 mM DTT, followed by immunoblotting as described above.

## In vitro ADP-ribosylation and detection by autoradiography
In vitro ADP-ribosylation reactions were performed as previously described (*Bonfiglio et al., 2017a*; *Palazzo et al., 2017b*). Reactions were stopped by Olaparib (2 μM final concentration) and then incubated with or without 1 M $NH_2OH$ for 3 hr before being detected by autoradiography. The molarity of HPF1 proteins used in the reactions were 1 μM, PARP1 was 0.1 μM and PARP1 E988Q 4 μM. The synthetic H3 peptide substrates were 3 μg per condition.

## Acknowledgements

Ahel laboratory is funded by the Wellcome Trust (grant 101794), Cancer Research UK (grant C35050/A22284), and the European Research Council (grant 281739). The work in Matic laboratory was funded by the Deutsche Forschungsgemeinschaft (Cellular Stress Responses in Aging-Associated Diseases) (grant EXC 229 to IM) and the European Union's Horizon 2020 research and innovation program (Marie Skłodowska-Curie grant agreement 657501 to IM). Luca Palazzo received a fellowship from the Italian Foundation for Cancer Research (FIRC, Milan, Italy; grant 14895). We are grateful to Ian Gibbs-Seymour for providing cell lines and advice, Juan-José Bonfiglio for discussions and Kai Heydenreich for help with cell culture experiments and western blotting. We would like to thank Johannes Rack and Edward Bartlett for the helpful comments on the manuscript.

## Additional information

### Funding

| Funder | Grant reference number | Author |
| --- | --- | --- |
| Associazione Italiana per la Ricerca sul Cancro | 14895 | Luca Palazzo |
| Cancer Research UK | C35050/A22284 | Ivan Matic |
| Horizon 2020 Framework Programme | 657501 | Ivan Matic |
| Wellcome Trust | 101794 | Ivan Ahel |
| Deutsche Forschungsgemeinschaft | EXC 229 | Ivan Ahel |
| Horizon 2020 Framework Programme | 281739 | Ivan Ahel |

The funders had no role in study design, data collection and interpretation, or the decision to submit the work for publication.

### Author contributions

Luca Palazzo, Orsolya Leidecker, Formal analysis, Investigation, Writing—original draft, Writing—review and editing; Evgeniia Prokhorova, Formal analysis, Investigation, Writing—review and editing; Helen Dauben, Formal analysis, Investigation; Ivan Matic, Formal analysis, Supervision, Project administration, Writing—review and editing; Ivan Ahel, Formal analysis, Supervision, Funding acquisition, Project administration, Writing—review and editing

### Author ORCIDs

Luca Palazzo http://orcid.org/0000-0002-5556-5549
Orsolya Leidecker http://orcid.org/0000-0001-5315-014X
Evgeniia Prokhorova http://orcid.org/0000-0002-5467-5586
Ivan Matic http://orcid.org/0000-0003-0170-7991
Ivan Ahel http://orcid.org/0000-0002-9446-3756

### Decision letter and Author response

Decision letter https://doi.org/10.7554/eLife.34334.010
Author response https://doi.org/10.7554/eLife.34334.011

## Additional files

### Supplementary files

• Transparent reporting form
DOI: https://doi.org/10.7554/eLife.34334.006

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
