## [Decision Letter]

Thank you for submitting your article "Serine is the major residue for ADP-ribosylation upon DNA damage" for consideration by *eLife*. Your article has been reviewed by three peer reviewers, and the evaluation has been overseen by a Reviewing Editor and Ivan Dikic as the Senior Editor. The reviewers have opted to remain anonymous.

The reviewers have discussed the reviews with one another and the Reviewing Editor has drafted this decision to help you prepare a revised submission.

This manuscript provides compelling data that supports ADP-ribosylation of serine residues as the major outcome of PARP1 activity during the DNA damage response. The paper will serve as a nice addition to the previously published paper identifying ARH3 as the enzyme that reverses the Ser-ADPr modification. The current study is important because it will highlight the prevalence of the serine modification during the DNA damage response, and could lead the field to re-focus on this type of modification, rather than just the Glu/Asp modification. In that regard, this is a valuable contribution to the literature and is worth publishing.

There are few experiments that need to be conducted.

1) The authors state that the major fraction of ADP-ribosylation occurs on serines after DNA damage. To substantiate the findings in Figure 3 that shows HPF1-dependent serine ADP-ribosylation of H2B, the authors should test whether knockout of HPF1 indeed abolishes serine 6 ADP ribosylation in vivo using MS. To what extent does Glu/Asp become the main ADP-ribose acceptors in the absence of HPF1? H2B does not appear to become ADP-ribosylated at all in the absence of HPF1, while PARP1 does. Is there any unbiased study addressing this switch in target residues?

2) Is serine-ADP ribosylation of other histones or additional main targets previously reported by authors such as HMGA1, HMGB1, HMGN1, NPM1, and TMA7 also dependent on HPF1. These experiments should be done in order to demonstrate that serine ADP ribosylation is the major fraction of ADP ribosylation after DNA damage.

3) What are the 'readers' of Ser-ADPr? This is referenced in the penultimate paragraph of the Results and Discussion section, but it is not clear what proteins carry out the specific reading of Ser-ADPr. A reference would be helpful here. The "erasers" of Ser-ADPr are more clear since the ARH3 enzyme is implied. It would be great to use ARH3 enzyme (as has been done in the previous *eLife* paper) to validate ADP-ribosylation on serine next to hydroxylamine (validating ADPr on Glu/Asp) in Figure 2.

4) The use of wild-type and S6A histone H2B is a nice example of specific ADPr modification, and there are not many examples in the literature specific ADPr sites confirmed in vivo. Are there other examples (e.g. other histones) that could be included in this study to solidify the technique and add important confirmed sites to the literature?

5) It is interesting to see that the level of basal/untreated ADP-ribosylation is often (at least on the used cell extracts) more than the levels detected upon recovery from DNA damage. Could the authors comment on this phenomenon? Also, while the study focuses on DNA damage-induced ADP-ribosylation, are you able to address which residues are predominantly modified in the untreated cells?

In summary, all reviewers were very favorable and kindly ask for the 4-5 points raised to be addressed.

---

## [Author Response]

There are few experiments that need to be conducted.1) The authors state that the major fraction of ADP-ribosylation occurs on serines after DNA damage. To substantiate the findings in Figure 3 that shows HPF1-dependent serine ADP-ribosylation of H2B, the authors should test whether knockout of HPF1 indeed abolishes serine 6 ADP ribosylation in vivo using MS.

We already addressed this point in our previous publication (Bonfiglio et al., 2017a). We showed that the ADPr on Ser6 of H2B is dramatically reduced (∼200-fold) in cells lacking HPF1 (Bonfiglio et al., 2017a; Figure 1).

To what extent does Glu/Asp become the main ADP-ribose acceptors in the absence of HPF1? H2B does not appear to become ADP-ribosylated at all in the absence of HPF1, while PARP1 does. Is there any unbiased study addressing this switch in target residues?

There is no unbiased study yet that would investigate in cells the switch in target residues from serines to glutamate. However, in our previous publication (Bonfiglio et al., 2017a) we have performed an unbiased mass spectrometric analysis of in vitro ADPr reactions addressing the switch in target residues on PARP-1 itself. In the current manuscript, our data show that the majority of ADPr under both basal and DNA-damage stimulus happens on Ser residues and that most of Ser-ADPr is lost in the absence of HPF1 (new Figure 2). Without HPF1 PARP1 is automodified significantly on the acidic residues and the average chain length substantially increases, while we cannot detect significant glutamate ADPr on histone in the same conditions. This is not surprising, since in the absence of HPF1 histone ADPr is effectively lost (while PARP1 becomes hyper-PARylated), as previously shown by us (Gibbs-Seymour et al., 2016).

2) Is serine-ADP ribosylation of other histones or additional main targets previously reported by authors such as HMGA1, HMGB1, HMGN1, NPM1, and TMA7 also dependent on HPF1. These experiments should be done in order to demonstrate that serine ADP ribosylation is the major fraction of ADP ribosylation after DNA damage.

Our previously published in vitro data (Bonfiglio et al., 2017a) showed strict HPF1-dependence of Ser-ADPr for all the core histones, HMGA1 (S8 and S9) and HMGN1 (S7) by using high-resolution ETD MS/MS analysis. ADPr of those target proteins was HPF1-dependent. In the same publication, we also showed that Ser-ADPr is lost on H3, H4, H1 and PARP1 in HPF1 KO cells.

3) What are the 'readers' of Ser-ADPr? This is referenced in the penultimate paragraph of the Results and Discussion section, but it is not clear what proteins carry out the specific reading of Ser-ADPr. A reference would be helpful here. The "erasers" of Ser-ADPr are more clear since the ARH3 enzyme is implied. It would be great to use ARH3 enzyme (as has been done in the previous eLife paper) to validate ADP-ribosylation on serine next to hydroxylamine (validating ADPr on Glu/Asp) in Figure 2.

The “readers” of Ser-ADPr are not known at the moment (we are sorry if our text was misleading, we have rewritten it) and this is another exciting area to explore in the future.

We have performed the requested experiment (see Author response image 1) and we noted that ARH3 removes the ADPr signal from both the wildtype and the HPF1 KO samples. This is not surprising, considering that it is well known that ARH3 removes poly-ADPr chains and that the absence of HPF1 leads to hyperPARylation (formation of very long chains of ADPr largely on PARP-1 itself), as shown by us previously (Gibbs-Seymour et al., 2016). In fact, treatment with PARG, which removes polyADPr without removing the terminal unit of ADPr, gives identical result in HPF1 KO samples (last lane). The complete loss of the ADPr in the HPF1 KO sample with either PARG or ARH3 might be surprising in the first instance. However, it is important to keep in mind that ADPr polymers can be extremely long (up to 200 units). Thus, by assuming that the average ADPr polymer contains 100 units of ADPr, the amount of ADPr that remains after the treatment with either ARH3 or PARG would be two orders of magnitude lower. Based on our experience with the anti-pan reagent, which is not very sensitive, it does not surprise us that this amount of ADPr does not give any detectable signal. In control cells, however, the amount of mono-ADPr seems to be a lot higher (with many more different sites modified), as supported by the PARG treatment experiment, therefore the ADPr signal is still detected.

We have performed this experiment for the reviewers, but we have decided not to include it in the manuscript, as the interpretation of this blot is very complicated (especially for non-specialist readers) and can, therefore, lead to misunderstanding of our message.

4) The use of wild-type and S6A histone H2B is a nice example of specific ADPr modification, and there are not many examples in the literature specific ADPr sites confirmed in vivo. Are there other examples (e.g. other histones) that could be included in this study to solidify the technique and add important confirmed sites to the literature?

We have now used the same technique to confirm the in vivo Ser-ADPr sites for H3 which worked perfectly (new Figure 4). The results fully support the previous data, and demonstrate that most of the main acceptor sites of ADPr on H3 is on Ser10 position and that there is also some ADPr targeting Ser28 (new Figure 4). Ser10 and Ser28 are the only ADP-ribosylation sites on histone H3 identified in our previous study (Leidecker et al., 2016) and the mutation of both residues completely abolished ADPr of histone H3 (new Figure 4). In addition, this result shows that our technique will be very useful for the confirmation of in vivo ADPr sites for different proteins.

5) It is interesting to see that the level of basal/untreated ADP-ribosylation is often (at least on the used cell extracts) more than the levels detected upon recovery from DNA damage. Could the authors comment on this phenomenon?

The cells not treated with the exogenous damage still experience damage from the endogenous sources including replication related stimuli. The fact that ADPr signal is often higher in the untreated samples than in the post-treated samples is likely due to upregulation of hydrolases in response to high doses of DNA damaging agents (as for example documented for PARG, which actively translocates to the nucleus upon DNA damage).

Also, while the study focuses on DNA damage-induced ADP-ribosylation, are you able to address which residues are predominantly modified in the untreated cells?

We have performed the experiment and as shown in new Figure 2, serine residues are the residues predominantly modified even under basal condition.